# Protective Characteristics of TiO$_2$ Sol-Gel Layer Deposited on Zn-Ni or Zn-Co Substrates

**Nelly Boshkova** [1], **Irina Stambolova** [2], **Daniela Stoyanova** [2], **Silviya Simeonova** [3], **Nikolay Grozev** [3], **Georgi Avdeev** [1], **Maria Shipochka** [2], **Ognian Dimitrov** [4], **Vasil Bachvarov** [1], **Miglena Peshova** [1] and **Nikolai Boshkov** [1,*]

1   Institute of Physical Chemistry "R. Kaishev", Bulgarian Academy of Sciences (BAS), 1113 Sofia, Bulgaria
2   Institute of General and Inorganic Chemistry, Bulgarian Academy of Sciences, 1113 Sofia, Bulgaria
3   Faculty of Chemistry and Pharmacy, Sofia University, 1164 Sofia, Bulgaria
4   Institute of Electrochemistry and Energy Sources, Bulgarian Academy of Sciences, 1113 Sofia, Bulgaria
*   Correspondence: nboshkov@ipc.bas.bg

**Abstract:** This study aimed to present the differences in the corrosion properties and protective ability of two bi-layer systems obtained on low-carbon steel in a model corrosive medium of 5% NaCl solution. These newly developed systems consist of Zn-Co (3 wt.%) or Zn-Ni (10 wt.%) alloy coatings as under-layers and a very thin TiO$_2$ sol-gel film as a top-layer. Scanning electron microscopy (SEM) is used for characterization of the surface morphology of the samples indicating that some quantitative differences appear as a result of the different composition of both zinc alloys. Surface topography is investigated by means of atomic force microscopy (AFM), and the hydrophobic properties are studied by contact angle (CA) measurements. These investigations demonstrate that both sample types possess grain nanometric surface morphology and that the contact angle decreases very slightly. X-ray diffraction (XRD) and X-ray photoelectron spectroscopy (XPS) are used for characterization of the chemical composition and electronic structure of the samples. The roughness R$_q$ of the Zn-Ni/TiO$_2$ is 49.5 nm, while for Zn-Co/TiO$_2$, the R$_q$ value is 53.4 nm. The water contact angels are 93.2 and 95.5 for the Zn-Ni/TiO$_2$ and Zn-Co/TiO$_2$ systems, respectively. These investigations also show that the co-deposition of Zn and Ni forms a coating consisting entirely of Ni$_2$Zn$_{11}$, while the other alloy contains Zn, Co and the intermetallic compound CoZn$_{13}$. The corrosion resistance and protective ability are estimated by potentiodynamic polarization (PDP) curves, as well as polarization resistance (R$_p$) measurements for a prolonged test period (35 days). The results obtained are compared with the corrosion characteristics of ordinary zinc coating with an equal thickness. The experimental data presents the positive influence of the newly developed systems on the enhanced protective properties of low-carbon steel in a test environment causing a localized corrosion—lower corrosion current density of about one magnitude of order (~$10^{-6}$ A.cm$^{-2}$ for both systems and ~$10^{-5}$ A.cm$^{-2}$ for Zn) and an enhanced protective ability after 35 days (~10,000–17,000 ohms for the systems and ~900 ohms for Zn).

**Keywords:** corrosion; zinc; Zn-Ni alloy; Zn-Co alloy; TiO$_2$ sol-gel layer

## 1. Introduction

Corrosion of metallic materials is a global problem due to the wide industrial and civil application of these materials—for example, for transport, buildings, bridges and other architectural and infrastructural purposes [1,2]. As a result of the interaction with the aggressive surrounding media, significant financial and materials losses appear often, accompanied with a negative influence on human health and the environment.

The operational conditions requirement determines the protection method, for example: galvanizing ("hot dip" technique or by means of electrodeposition); application of inhibitors; different types of barrier coatings; cathodic protection; conversion films and etc.

Generally, the galvanizing process enhances the corrosion resistance of the low-carbon and low-alloyed steels, mainly due to the protective barrier effect of the galvanic layer [3]. As a result, the zinc electrodeposition process is widely used in different industries: automotive, electricity, food, piping, building, shipping and etc. [4]. A disadvantage is the insufficient protection of the zinc corrosion products in very aggressive media.

To further protect the zinc-coated steel against corrosion, surface modification is generally adopted. The good corrosion resistance of galvanized steel could be improved by alloying Zn with selected metals such as Co, Ni, Mn, Fe and etc. [5–7]. Another approach that can be used to reduce the corrosion processes is covering the galvanized steel with barrier oxide nanocoatings such as $TiO_2$, $ZrO_2$, $CeO_2$ and etc. The good wear and anticorrosion properties of these nanocoatings are due to the high density of the grain boundaries and very fine particles. As a result, they can restrict the diffusion process of the corrosive agents deeply inside to the metal substrate. Among them, $TiO_2$ film is a technologically important material that is widely used in many areas: solar cells, photocatalytic purification of water, self-cleaning windows and etc. During the past decades, $TiO_2$ was a very attractive object of investigation due to its low price, chemical stability, and excellent electrical, chemical, and optical features, and especially due to its corrosion resistance properties. Sol-gel is popular, low-cost method for the preparation of various multicomponent powders and films. Lower processing temperatures and high chemical homogeneity results in better controlling of the final physical and chemical properties of the materials [8,9]. In comparison to some other physical and chemical films deposition methods, sol-gel ensures the preparation of uniform nano-sized coatings with a complex shape, demonstrating good adhesion to different substrates.

The scientific data concerning the application of $TiO_2$ coatings as a barrier against corrosion of low-carbon steel are relatively scarce. Nanostructured $TiO_2$ protective thin layers were deposited on stainless steel by spray pyrolysis, a sol-gel method. [10,11]. Some researchers have reported higher corrosion resistance of galvanized steel using modified nano-sized $TiO_2$-particle films [12]. Romero et al. have successfully applied spray pyrolysis for deposition of various inorganic oxide films such as NiO, MgO [13], $ZrO_2$ [14] or ZnO [15], having an aim to improve the protective characteristics of the galvanized steel. The effective protection of galvanized steel in comparison to the uncoated galvanized one has also been proved by the application of selected inorganic oxide coatings [16–18].

Our previous experiments with $TiO_2$ films deposited on electrogalvanized carbon steel have revealed that after the thermal treatment, the films possess very deep cracks and in some areas they are almost delaminated from the metal substrate. This forced us to look for another experimental approach to improve the adhesion of the oxide layer. According to some previous studies [5–7], the alloying of the zinc with other metals (Co, Mn, Ni, etc.) leads to better well-expressed corrosion resistance in chloride containing test medium due to the appearance of an additional barrier layer of the newly formed corrosive products with a low product of solubility value.

To the best of our knowledge, the available scientific literature does not present sufficient data about the corrosion properties of protective systems on low-carbon steel consisting of a zinc alloy (Zn-Ni or Zn-Co, respectively) additionally covered by a sol-gel titanium dioxide layer.

The aim of the present investigation is to characterize the corrosion behavior of two novel bi-layer systems containing Zn-Co (3 wt.%) or Zn-Ni (10 wt.%) as under-layers and an additional $TiO_2$ sol-gel coating as a top-layer in a model test medium of 5% NaCl and to compare it with ordinary zinc coating. An additional aim is to evaluate the impact of the nature of the under-layer.

## 2. Materials and Methods

### 2.1. Sample Types

Low-carbon steel plates (Metalsnab, Sofia, Bulgaria) with a size of 30 mm × 10 mm × 1 mm were used as substrates. Two bi-layer systems were obtained and investigated:

-   System A: Zn-Ni (10 wt.%)—under-layer; $TiO_2$—top-layer;
-   System B: Zn-Co (3 wt.%)—under-layer; $TiO_2$—top-layer.

Both zinc-based alloys and ordinary zinc (for comparison) have been electrodeposited with equal thickness of about 12 μm. The alloy coating Zn-Co (3 wt.%) was obtained from an electrolytic solution containing 100 g/L $ZnSO_4 \cdot 7H_2O$ (Valerus, Sofia, Bulgaria), 120 g/L $CoSO_4 \cdot 7H_2O$ (Valerus, Sofia, Bulgaria), 30 g/L $NH_4Cl$ (Valerus, Sofia, Bulgaria) and 25 g/L $H_3BO_3$ (Valerus, Sofia, Bulgaria). The pH value of the electrolyte was between 3.0–4.0, and the cathodic current density was 2 $A/dm^2$. In addition, soluble zinc anodes (Valerus, Sofia, Bulgaria) and additives ZC-1 (wetting agent—20 mL/L, IPC-BAS, Sofia, Bulgaria) and ZC-2 (brightener—2 mL/L, IPC-BAS, Sofia, Bulgaria) were used [5,6]. The other alloy (Zn-Ni; 10 wt.%) was obtained in a thermostatic electrolytic cell with circulation from an electrolyte with a composition of 100 g/L $NiSO_4.7H_2O$ (Valerus, Sofia, Bulgaria), 100 g/L $NiCl_2 \cdot 6H_2O$ (Valerus, Sofia, Bulgaria) 30 g/L $ZnCl_2$ (Valerus, Sofia, Bulgaria), and 10 g/L β-alanine (Valerus, Sofia, Bulgaria). The pH value was 4. The electrodeposition process was realized at a cathodic current density of 2 $A/dm^2$, a temperature of 40 °C, and with non-soluble Ti-Pt networks as anodes [7].

Ordinary zinc coating was electrodeposited from solution with a composition of 150 g/L $ZnSO_4 \cdot 7H_2O$, 30 g/L $NH_4Cl$, and 30 g/L $H_3BO_3$ at the following conditions: pH 4.5–5.0, cathodic current density 2 $A/dm^2$, soluble zinc anodes, and 2 additives: wetting agent (AZ1) and brightener (AZ2) [5,6].

Thereafter, the $TiO_2$ sol-gel layer was deposited on the zinc alloy coated steel substrates according to the procedure described below. Titania-based nanosized coatings were prepared from titanium tetrabutoxide (TB, Alfa Aesar, Haverhill, MA, USA) dissolved in isopropanol and acetylacetone (0.4 M/L, Sigma Aldrich, Saint Louis, MO, USA). Afterwards, small quantity of polyoxyethylene (20) sorbitan monooleate (Tween 80, Sigma Aldrich, Saint Lous, MO, USA) was added under vigorous stirring for 1 h. As well-known, Tween 80 is a biodegradable surface modifier and has a remarkable effect on the reduction of the crystalline sizes. The substrates were dipped in the precursor solution and were withdrawn at a rate of 30 mm/min. After each deposition, the samples were dried at 50 °C for 30 min, and thereafter, treated consecutively at 180 °C for 1 h and at 380 °C for 1 h (velocity rate 3°/min). The dipping-drying cycle was repeated three times.

### 2.2. SEM/EDX Investigations

The surface morphology of both investigated systems was checked with a scanning electron microscopy unit (Oxford Instruments, Oxford, UK). EDX analyses of the systems A and B were carried out in four different points and revealed similar chemical composition.

### 2.3. AFM Studies

The surface topography and morphology were characterized by the application of atomic force microscopy (NanoScopeV system, Bruker Ltd., Bremen, Germany) operating in tapping mode in air at room temperature. Silicon cantilevers (Tap 300 Al-G, Budget Sensors, Innovative Solutions Ltd., Sofia, Bulgaria) with 30 nm thick aluminum reflex coatings were used. The cantilever force constant was in the range 40 N/m, and the resonance frequency was 300 kHz. The scanning rate was set at 1 Hz. The roughness analysis (using Nanoscope software, Bruker Inc., Birrica, MA, USA) gives the value $R_a$, which is an arithmetic average of the absolute values of the surface height deviations measured from the mean plane, while $R_q$ is the root mean square average of height deviations taken from the mean image data plane.

### 2.4. Contact Angle Measurements

The investigations were performed with a Ramé-Hart automated goniometer model 290 with DROP image advanced v2.4 (Succasunna, NJ, USA) at room temperature. Small water drops of 2–5 μL were formed and deposited with a Ramé-Hart automatic dispensing system. The contact angles of 10 consecutive drops positioned at random locations of the

samples were measured. A mean angle and a mean error were taken from them. Wettability of solid surface is determined through contact angle. Young's equation defines the contact angle 'θ' by analyzing the forces acting on a fluid droplet resting on a solid surface. The contact angle 'θ' is the angle formed by a liquid at the three-phase boundary where the liquid, gas and solid intersect. According to Young's equation, the relationship between these four parameters is

$$\gamma_{SG} = \gamma_{SL} + \gamma_{LG} \times \cos\theta \tag{1}$$

*γ—surface tension (energy) of solid-gas interface*
*SG— solid-gaseous, SL— solid-liquid, LG— liquid-gaseous*

A contact angle of less than 90° indicates that wetting of the surface is favorable. Otherwise, the wettability is unfavorable [19].

### 2.5. Chemical and Phase Composition

X-ray photoelectron spectroscopy (XPS) was used for the investigation of the chemical composition and electronic structure of the films. AXIS Supra electron-spectrometer (Kratos Analitycal Ltd., Manchester, UK) using achromatic AlKα radiation with a photon energy of 1486.6 eV and charge neutralization system was applied for this purpose. The binding energies (BE) were determined with an accuracy of ±0.1 eV, using the C1s line at 284.6 eV (adsorbed hydrocarbons). The chemical composition in the depth of the films was determined by monitoring the areas and binding energies of C1s, O1s, and Ti2p photoelectron peaks. Using the commercial data-processing software of Kratos Analytical Ltd., the concentrations of the different chemical elements (in atomic %) were calculated by normalizing the areas of the photoelectron peaks to their relative sensitivity factors.

### 2.6. XRD Analyses

The phase composition of the samples was studied with X-ray diffraction analysis by using an X-ray diffractometer with CuKα radiation and a generator voltage of 40 kV. The diffractometer was equipped with a PW1830 generator and a PW1050 goniometer manufactured by Philips. The experiment was carried out in the following configuration: an X-ray tube with a copper anode, a scintillation detector and a diffracted radiation monochromator. Data were acquired in the angular range of 5–90° 2 theta with a step of 0.05° 2 theta and exposure 3 s. The HighScore Plus 3.0 program (Malvern Analytical, Almelo, The Nederlands) and Inorganic Crystal Structure Database (ICSD) were used for phase analysis.

### 2.7. Electrochemical Tests

Two well-known electrochemical methods were applied to estimate the corrosion behavior of the sample systems: potentiodynamic (PDP) polarisation curves and polarisation resistance ($R_p$) measurements. Detailed description of these techniques and devices have been reported by the authors previously [5–7,16]. The investigations were realized in a three-electrode electrochemical cell having a volume of 300 mL. The saturated calomel electrode was the reference electrode, and the platinum wire was the counter one.

### 2.8. Corrosive Medium and Reproducibility

All electrochemical tests have been carried out in a model corrosive medium of 5% NaCl solution at ambient temperature. The reproducibility of the tests was an average of five samples per sample type.

## 3. Results

### 3.1. SEM Investigations

The surface morphology of both investigated systems is shown in Figure 1. It is obvious that some qualitative differences appear, with the latter being most probably a result of the differrent compositions of the electrodeposited zinc alloys—Zn-Ni and Zn-Co, respectively.

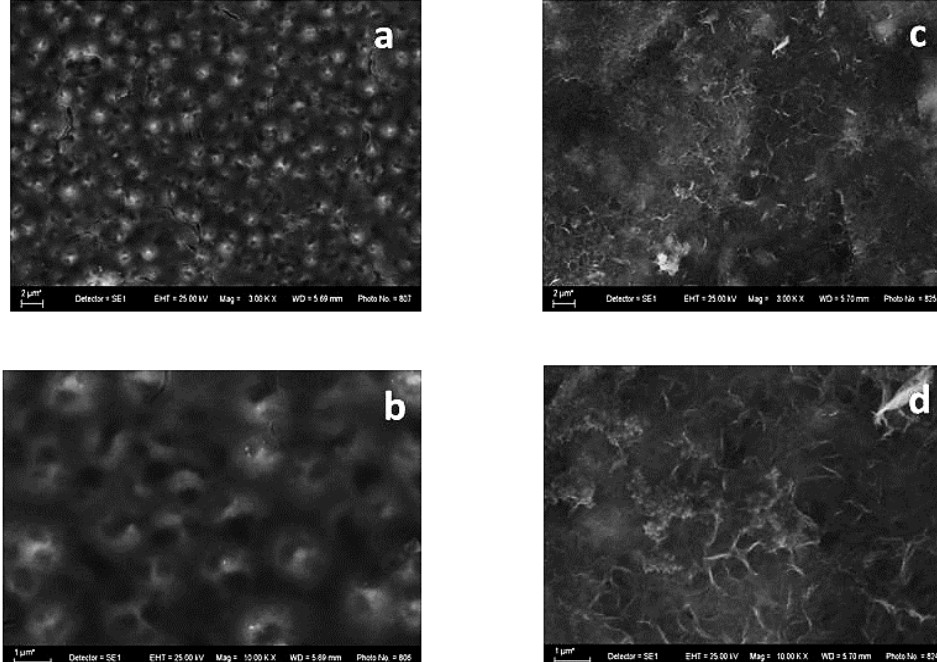

**Figure 1.** SEM images of System A—fresh sample (left): magnification 3000 (**a**) and 10,000 (**b**) and after corrosive treatment (right): magnification 3000 (**c**) and 10,000 (**d**).

The sample with Zn-Ni alloy as the underlayer (System A) demonstrates, in general, a smooth "bubble"-like surface (Figure 1a,b). Many spherical formations can be observed, with the latter having a visible opening/hole at the upper end. This implies the presence of empty internal volumes that could be filled with a corrosive environment, leading to an accelerated corrosion attack deeply inside.

Figure 1c,d shows that the films are covered with newly appeared corrosive products as a result of the immersion in the test medium. The layer of corrosion products is relatively even and random.

Zn, O, C, Ni and Ti (originated from $TiO_2$ sol-gel coating) were registered in the fresh sample of System A according to the EDX analyses (Figure 2) and after corrosive treatment (Figure 3).

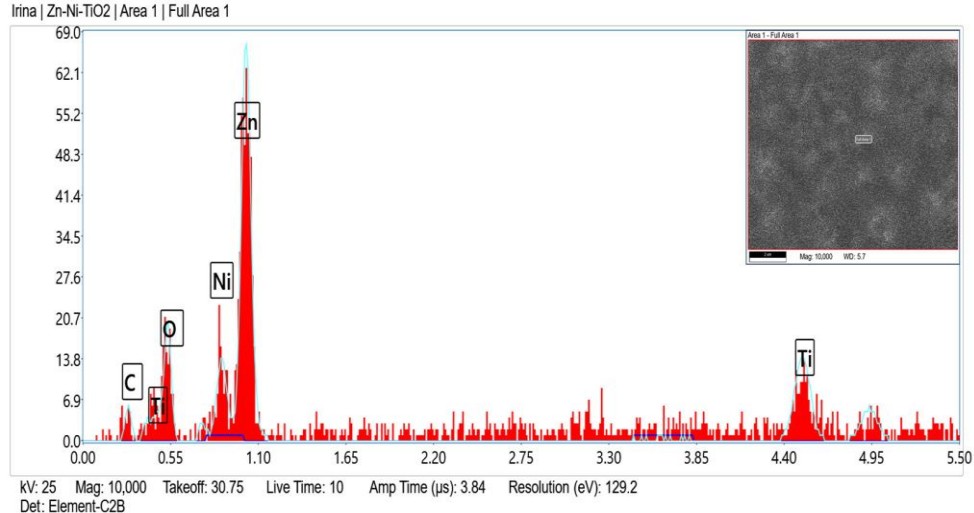

**Figure 2.** EDX spectrum of System A fresh sample.

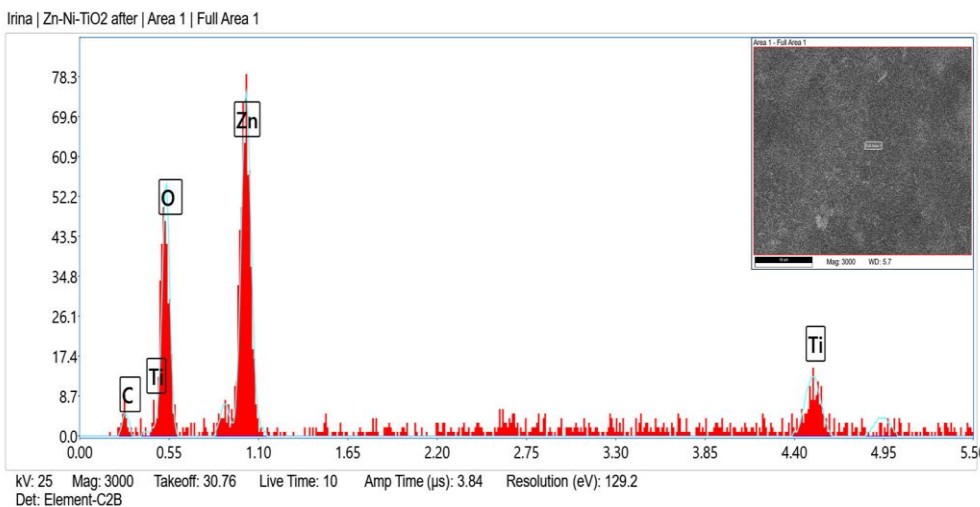

**Figure 3.** EDX spectrum of System A after corrosive attack.

The SEM photographs of the System B reveal a relatively smooth surface with small pores and holes (Figure 4a,b). As in the case of System A, after the corrosion treatment, some NaCl crystals are visible, and the surface morphology of System B remains relatively even. Contrary to the previous case, it seems that some parts of the coating are without corrosion damages (Figure 4c).

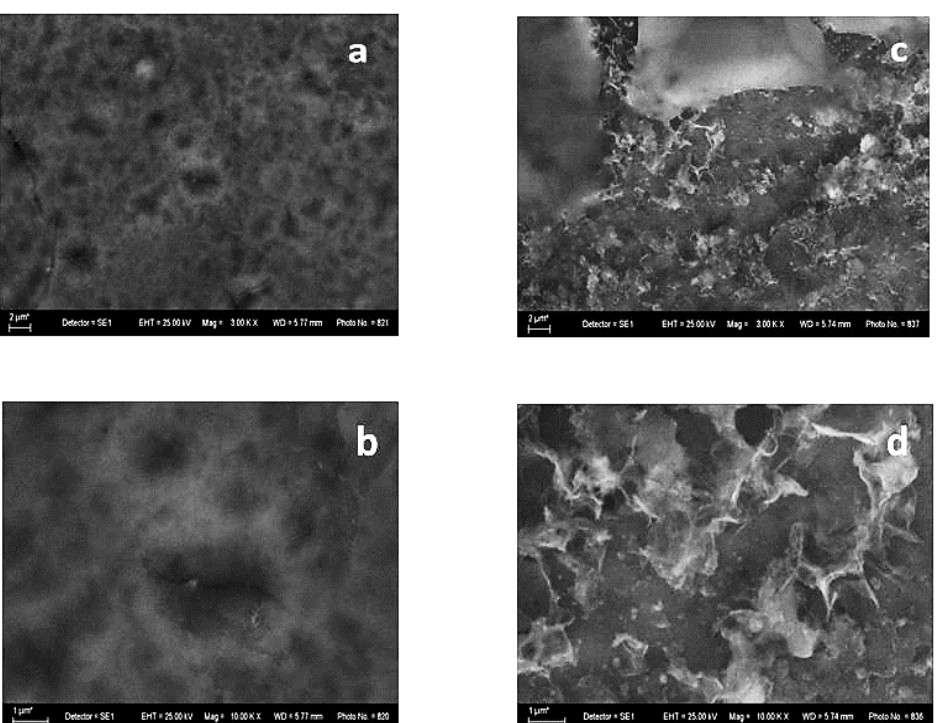

**Figure 4.** SEM images of System B—fresh sample (left): magnification 3000 (**a**) and 10,000 (**b**) and after corrosive treatment (right): magnification 3000 (**c**) and 10,000 (**d**).

Figures 5 and 6 present the EDX spectra of System B, which are to a certain degree similar to the corresponding EDX spectra of System A. The presence of the chlorine peak originated from the corrosive medium.

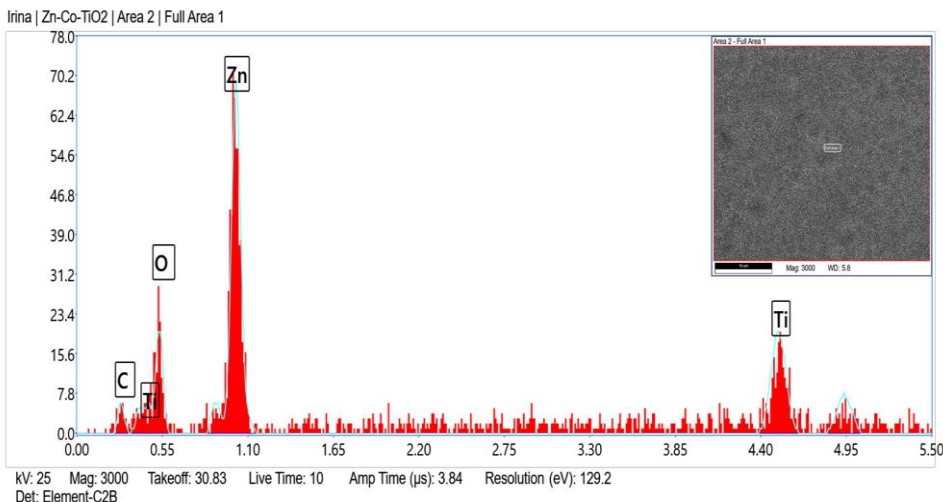

**Figure 5.** EDX spectrum of System B fresh sample.

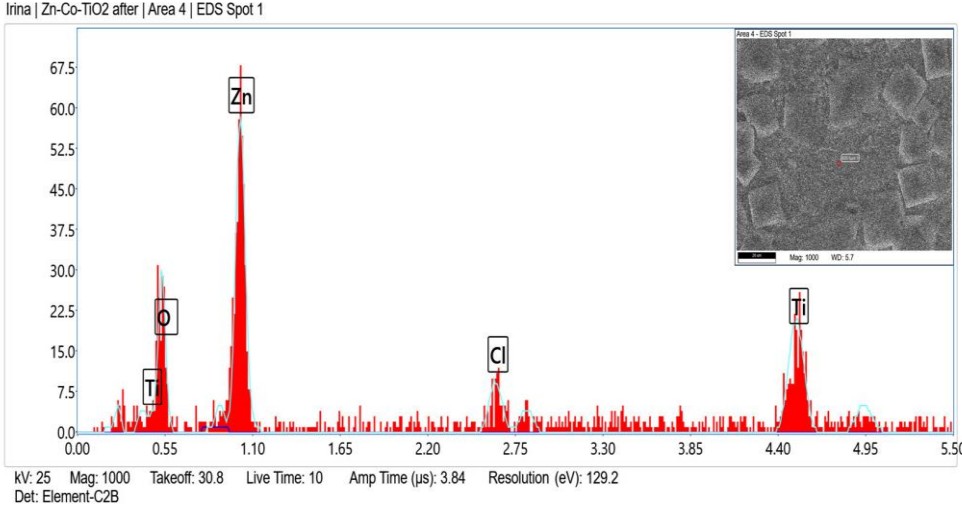

**Figure 6.** EDX spectrum of System B after corrosive attack.

Based on the SEM and EDX analyses, both systems possess the following characteristics (features):

(i) presence of Ti peak in EDX after the corrosion test;
(ii) the surface morphology of $TiO_2$ coating remains almost unchanged, and any signs of corrosion such as cracks, craters, pits and etc. are not observable.

### 3.2. Surface Topography by AFM Studies

The surface topography of Zn-Ni and Zn-Co alloys as well as Systems A and B, respectively, were studied by means of atomic force microscopy.

The AFM images of both the Zn-Ni alloy sample and System A were compared and presented in Figure 7. The morphology of the Zn-Ni alloy with a scanning area of $10 \times 10$ µm$^2$ (Figure 7 *left*) is rough with the presence of spherical "cluster" structures, in comparison with the morphology of the System A with the same scan area (Figure 7 *right*), which is more smooth with a homogeneous structure. Similarly, the AFM images of both types of Zn-Co alloy and System B samples were compared and presented in Figure 8. The morphology of the Zn-Co alloy sample with a scanning area of $10 \times 10$ µm$^2$ (Figure 8 *left*) demonstrate a smooth surface. In the presence of $TiO_2$, the morphology of System B sample changes with the observed smaller structures clustering into larger regions (Figure 8 *right*). Generally, these studies are very close to the SEM investigations presented above.

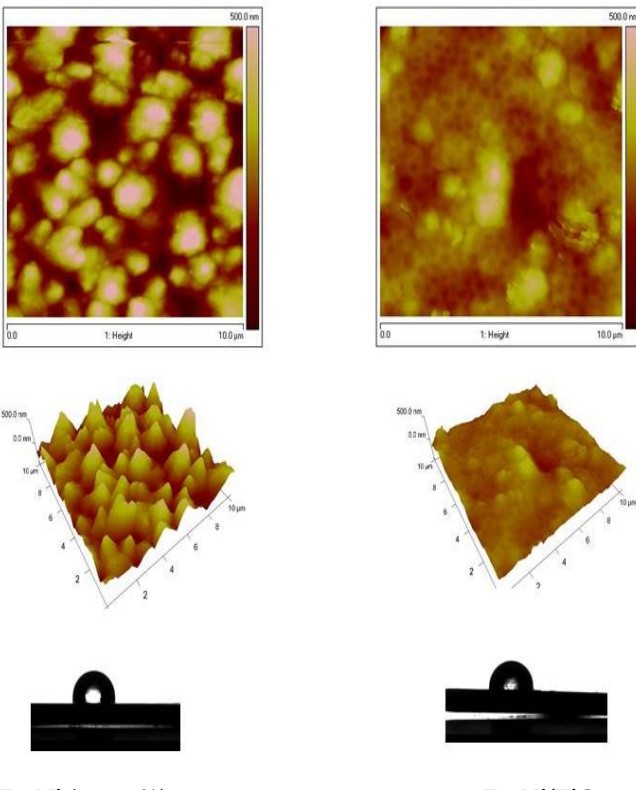

**Zn-Ni (10 wt.%)**  **Zn-Ni/TiO₂**

**Figure 7.** AFM topography of Zn-Ni alloy and Zn-Ni/TiO$_2$ (System A) samples: 2D images, 3D images and water contact angle.

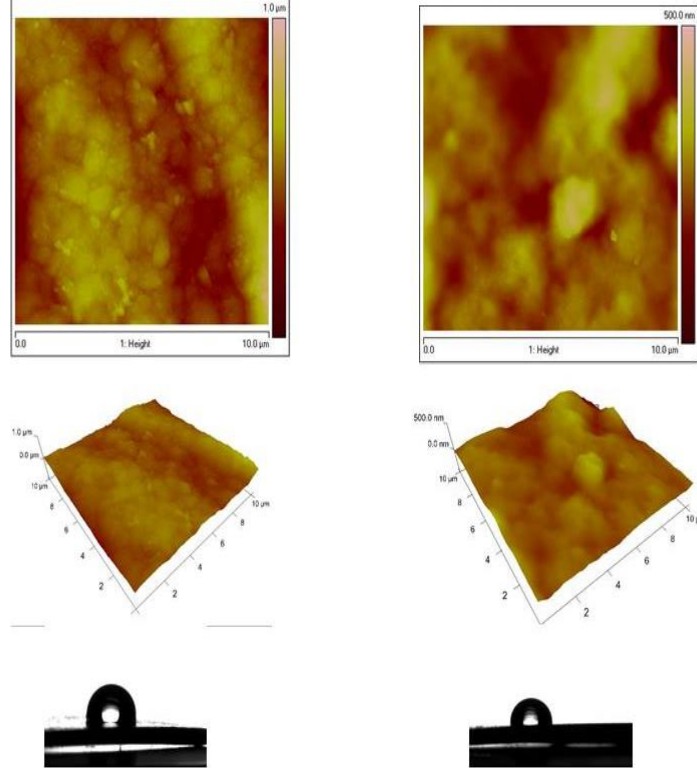

**Figure 8.** AFM topography of Zn-Co alloy and Zn-Co/TiO$_2$ (System B) samples: 2D images, 3D images and water contact angle.

For the samples that included Ni, the roughness $R_q$ of the Zn-Ni alloy sample is 125 nm, while the roughness of the System A is $R_q$ = 49.5 nm, i.e., it decreases 2.5 times. For the samples that included Co, the roughness $R_q$ of the Zn-Co alloy is 94.6 nm, while the roughness of the Zn-Co/TiO$_2$ is $R_q$ = 53.4 nm, i.e., it decreases by 2 times. Additional information is presented in Table 1 below.

**Table 1.** Roughness values $R_a$, $R_q$ and water droplet contact angle of the samples—Zn-Ni, Zn-Co and with addition of TiO$_2$—System A and System B, respectively.

| Samples | $R_a$, nm | $R_q$, nm | Contact Angle |
|---|---|---|---|
| Zn-Ni | 99.5 | 125 | 97.3 |
| System A | 36.7 | 49.5 | 93.2 |
| Zn-Co | 77.8 | 94.6 | 106.5 |
| System B | 41.1 | 53.4 | 95.5 |

It can be concluded that both types of samples (Zn-Ni and Zn-Co alloys), coated with TiO$_2$ film (System A and System B) possess grain nanometric surface morphology. After the deposition of the TiO$_2$ coating in both investigated systems, the water contact angles decreased very slightly (Table 1).

*3.3. Phase Composition*

The phase composition and structure of both systems, Zn-Ni/TiO$_2$ and Zn-Co/TiO$_2$, can be observed in Figure 9.

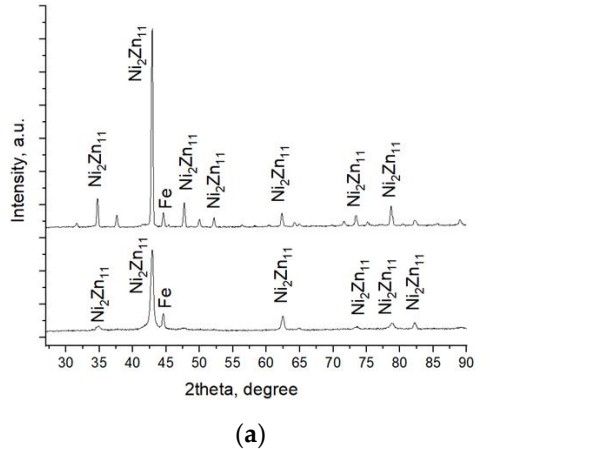

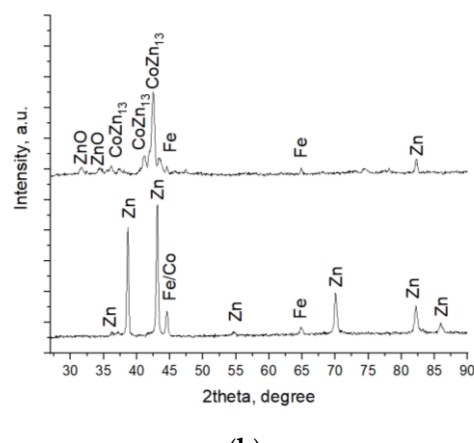

(**a**)  (**b**)

**Figure 9.** X-ray diffractograms of systems A (**a**) and B (**b**).

Co-deposition of zinc and nickel forms a coating consisting almost entirely of Ni$_2$Zn$_{11}$ (#PDF 01-072-2671). In addition, the peak of the iron substrate can also be observed (Figure 9a). The heating procedure after the TiO$_2$ deposition leads to a better crystallization of the already deposited galvanized Zn-Ni coating, and this statement is proved by the increased intensity of peaks in System A. No diffraction peaks of any TiO$_2$ crystallographic phases were observed.

As can be seen from Figure 9b, zinc and cobalt were co-deposited as an under-layer in the form of an alloy. Figure 9b shows the diffractograms of Zn-Co and System B. The X-ray pattern of the Zn-Co sample clearly proved the presence of the zinc phase Zn ICSD 98-005-3769 #PDF 01-071-4620; however, the strongest cobalt peak (#PDF 00-015-0806) practically coincides with that of the steel substrate (#PDF 00-006-0696). Therefore, its distribution is difficult to be strictly fixed in one phase. Most likely, it is also present as a separate phase and partially dissolved in zinc. The deposition process of TiO$_2$ coating requires heating of the sample, which leads to phase changes in the under-layer. As a result, a new intermetallic compound CoZn$_{13}$ (#00-029-0523) was registered. The latter is very similar in structure and properties to another intermetallic compound, CrZn$_{13}$,

which characterizes with well-expressed corrosion resistance [20]. A small amount of zinc is converted to zinc oxide (#PDF 01-071-3830). Similarly, as in System A, diffraction peaks of the most common TiO$_2$ crystallographic forms anatase and rutile in System B were not registered. This is probably related to the low heating temperature of the samples at which the titanium-oxide phase fails to crystallize and remains amorphous.

### 3.4. XPS Investigations

The surface composition and chemical state of the anticorrosive layers were investigated by XPS. XPS analysis noted peaks of O1s, C1s, Zn2p, Ti2p, Cl2p, Na1s, Fe2p and Ca2p on the surface of the films. The layers treated in 5% NaCl show different features than the untreated samples. The carbon spectra were deconvoluted by a Lorentzian–Gaussian curve fitting into several components. The first ones at ~283.0 eV are associated to C-Zn, C-Fe bonds. The following components correspond to adventitious carbon contamination at binding energies ~284.8 eV (C-C), ~286.3 eV (C-O-C) and ~288.6 eV (O-C=O) (Figure 10). Cl2p and Na1s peaks with binding energies at 198.5 eV and 1071.6 eV are recorded, which are attributed to the NaCl compound.

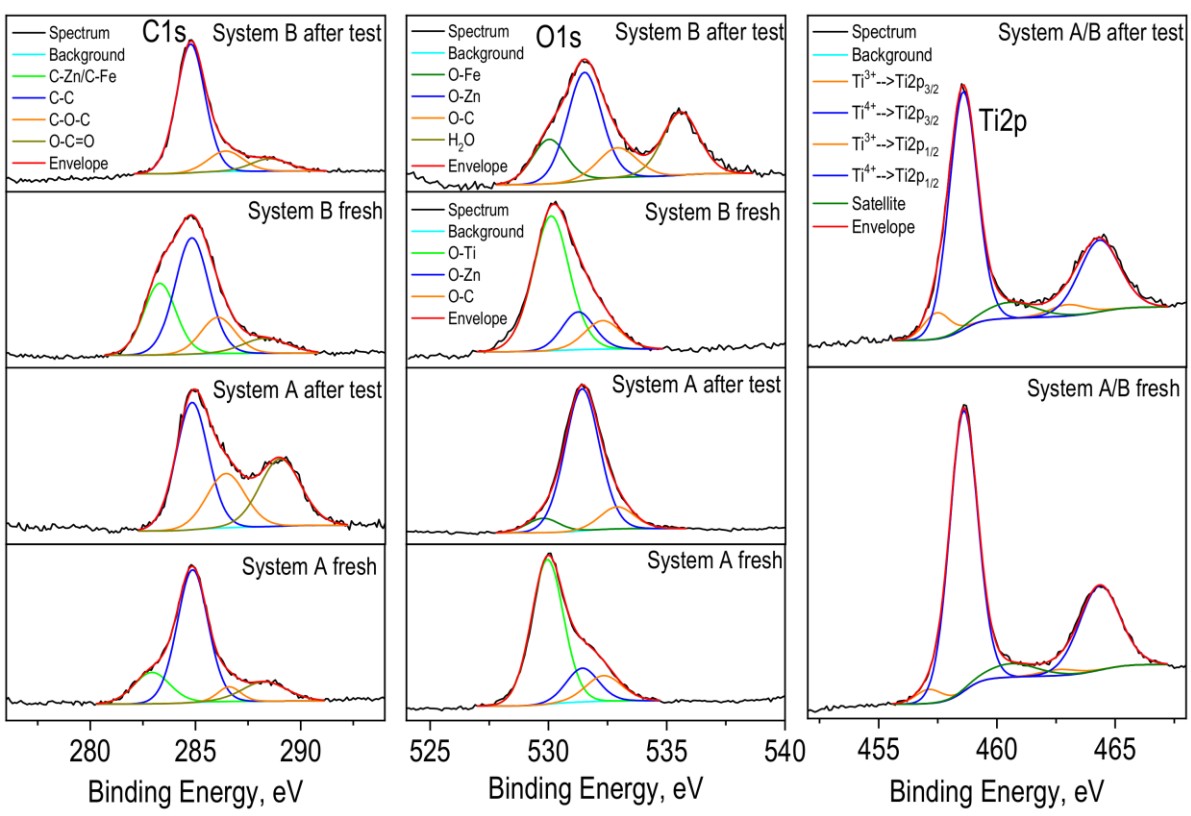

**Figure 10.** Deconvolution of C1s, O1s and Ti2p photoelectron spectra of the Systems A and B.

The O1s spectra show two mean peaks at 530.0 eV and 531.4 eV, which is assigned to lattice oxygen in TiO$_2$ (untreated layer), Fe$_2$O$_3$ (treated layer) and in ZnO, respectively. After treatment of the coatings in a corrosive medium, the peaks corresponding to ZnO dominates, while in untreated layers, the peak associated with TiO$_2$ is more intense. As well-known from [5,6] ZnO or Zn(OH)$_2$ is part of the compound zinc hydroxide chloride (Zn$_5$(OH)$_8$Cl$_2$.H$_2$O, ZHC), which appears as a main component on the corrosive-treated zinc in that medium. ZHC has a very low product of solubility value and ensures a better barrier effect and protective ability of the metal. A shoulder appears in higher binding energies at ~532.6 eV and is attributed to the C-O bond.

In the Zn-Co sample after the corrosion test, oxygen in the water molecule is also observed (Figure 10). The photoelectron spectra of Zn2p (not presented) show two peaks

with binding energies at ~1021.8 eV for $Zn2p_{3/2}$ and ~1044.9 eV for $Zn2p_{1/2}$. Observed peak positions and spin orbital splitting between peaks $2p_{3/2}$ and $2p_{1/2}$ of 23.1 eV are characteristics of ZnO. A shoulder in the lower binding energies is also observed in the spectra of zinc before corrosion tests, which is due to the Zn-C bond and is also confirmed by the spectrum of carbon (Figure 10).

The $Ti2p_{3/2}$ peaks have a maximum at 458.6 eV, which is typical for the $Ti^{4+}$ oxidation state. An asymmetry in the lower binding energies (~457.3 eV) of titanium peaks is seen, which corresponds to the $3^+$ oxidation state. Insignificant amounts of calcium and ferrum were also registered on the surface of the layers.

### 3.5. Electrochemical Investigations

The results from the potentiodynamic polarization curves of the samples in the model test solution of 5% NaCl is shown in Figure 11 and these of the polarization resistance in Figure 12.

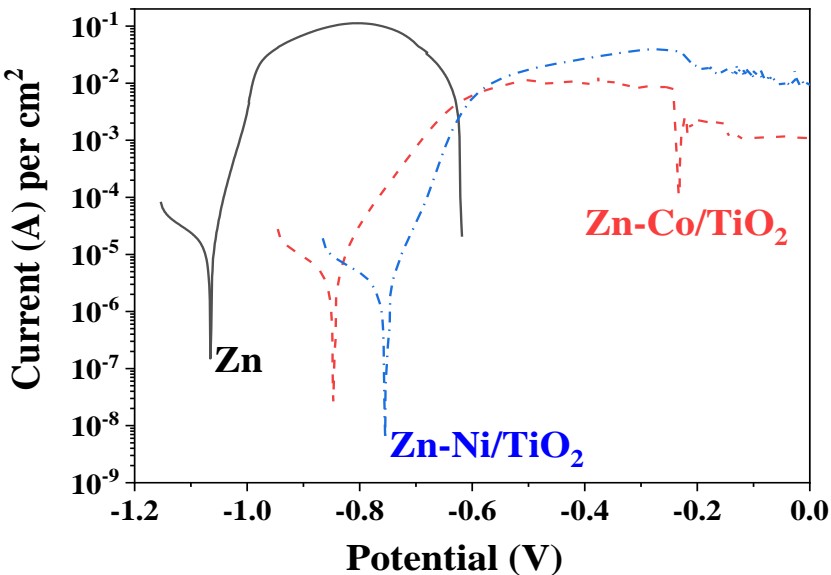

**Figure 11.** PDP curves of ordinary Zn, System A ($Zn-Ni/TiO_2$) and System B ($Zn-Co/TiO_2$).

It is evident that the corrosion potential of System A is the most positive one compared to System B and ordinary zinc. The corrosion current density of both bi-layer systems are relatively close and with approximately one order of magnitude lower compared to the zinc sample. The anodic curve of the ordinary zinc is the shortest one since the coating is practically fully dissolved at a potential zone of approximately −0.6 V, which is checked by the "naked eye". Both protective systems A and B demonstrate better corrosion resistance at conditions of external anodic polarization.

The anodic curve of the system $Zn-Ni/TiO_2$ shows a steeper slope, which is a sign for accelerated dissolution in the potential area right after the corrosion potential. The same sample also demonstrates higher anodic current density in the zone of the maximal anodic dissolution (−0.55 up to −0.25 V), which is approximately one order of magnitude higher compared to the $Zn-Co/TiO_2$ system. One reason for such behavior is most likely the relative surface inhomogeneity and the presence of some holes and pits in System A (see Figure 1), which could act as places for accelerated corrosion processes. Contrary to this, the $Zn-Co/TiO_2$ system has a slower anodic process and lower current density value in the anodic zone of maximal dissolution (−0.55 up to −0.25 V). Thereafter, this sample shows a trend for passivation, with the latter being weakly expressed ("pseudo-passivation").

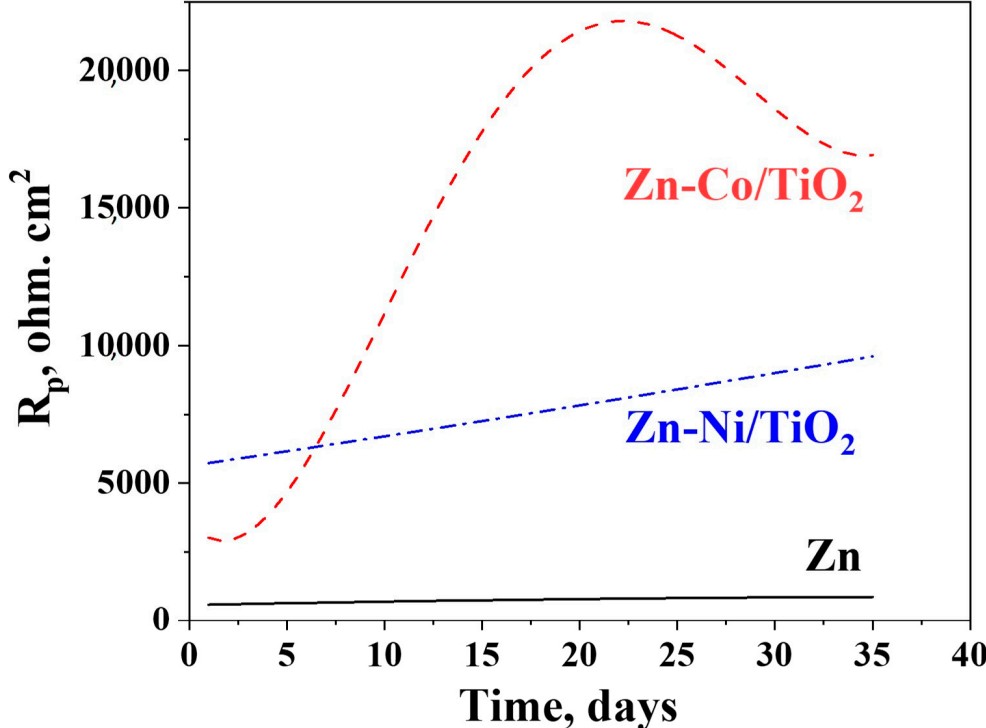

**Figure 12.** Polarization resistance measurements of ordinary Zn, System A and System B.

The most important electrochemical parameters are presented in Table 2.

**Table 2.** Some electrochemical parameters from PDP curves.

| No. | Sample | $I_{corr}$, $A.cm^{-2}$ | $E_{corr}$, V |
|-----|--------|-------------------------|---------------|
| 1 | Ordinary Zn | $1.8 \times 10^{-5}$ | −1.06 |
| 2 | Zn-Ni/TiO$_2$ | $2.4 \times 10^{-6}$ | −0.85 |
| 3 | Zn-Co/TiO$_2$ | $3.5 \times 10^{-6}$ | −0.75 |

The experimental data obtained for the polarization resistance of the investigated samples for a prolonged period of 35 days is presented in Figure 12.

It can be registered from the figure that the polarization resistance of the ordinary zinc is very low compared to both other investigated objects. It increases gradually in time, reaching about 1000 ohm.cm$^2$ at the end of the test period. The same parameter for the Zn-Ni/TiO$_2$ system also increases gradually up to about 10,000 ohm.cm$^2$, i.e., ten times higher than the zinc. The Zn-Co/TiO$_2$ system has a maximum in the Rp values at the 20th day, followed by a decrease in the corrosion resistance. However, it is obvious that this system demonstrates the best corrosion resistance and protective ability almost during the whole time (except the first 5–6 days). The reason for this observation seems to be to a great degree the surface morphology and topography (greater inhomogeneity) but also the nature and quantities of the newly appeared corrosion products.

XRD analyses have proved a lower crystallization degree of zinc-cobalt alloy under-layer in System B. The combination of the low crystallized under-layer with a fully amorphous TiO$_2$ coating leads to a structure with higher corrosion stability. Contrary to this, System A (ZnNi –TiO$_2$) exhibited a higher degree of crystallization of the Zn-Ni alloy, and despite the amorphous nature of the top-layer of the TiO$_2$ film, it is not so stable from the corrosion point of view compared to System B.

## 4. Discussion

The experimental methods for corrosion characterization used by the authors are accelerated (potentiodynamic polarization curves) and prolonged (polarization resistance measurements for 35 days of immersion of the samples). Both methods suggest the possibility to check the corrosion behavior for short- and long-time periods for better understanding of the nature of the tested samples. The other techniques used for the surface characterization of the investigated samples are very useful for creating a complete view of the newly developed materials and can also be applied when the objects are upgraded.

The data in the literature revealed that the presence of fully or partially amorphous structures increases the corrosion stability of zirconia and ceria oxide films [21,22]. The authors have explained this effect with the reduction of the diffusion of the ionic species. This could be due to the surface characteristics of the amorphous structure: a chemically homogeneous structure without any defects, which could initiate the corrosion process. The hydrophobic nature of the film's surface confirmed by the water contact angle above 90° also contributes to the enhanced protective characteristics.

Other reason for the increased corrosion resistance is the surface morphology and topography of both systems that strongly differ (see Figures 1, 4, 7 and 8). In the case of System A, practically the whole surface is covered with newly appeared corrosion products that protects, to a certain degree, the under-layer and the substrate. Due to the appearance of some holes, the corrosive medium can penetrate deeply inside the system, leading to an appearance of corrosive products (increasing the protective ability) and also to a gradual dissolution of the coating. In the case of System B, practically no holes/"bubbles" are present. Additionally, it is obvious that part of the top-coating (Figure 4c) remains, which is a sign of better corrosion resistance. A new compound is also registered in the case of System B ($CoZn_{13}$), which is investigated by other scientists [23], and the obtained results show its enhanced corrosion resistance. Additional information can be found in Table 3.

**Table 3.** Polarization resistance measurements after 35 days immersion in 5% NaCl.

| No. | Sample | $R_p$, Ohm.cm$^2$ (35 Days) |
| --- | --- | --- |
| 1 | Ordinary zinc | 885 |
| 2 | Zn-Ni/$TiO_2$ | 9684 |
| 3 | Zn-Co/$TiO_2$ | 16,918 |

## 5. Conclusions

Optimum conditions were found for the preparation of corrosion resistant bi-layer systems based on selected zinc alloys (Zn-Co and Zn-Ni) for enhanced corrosion resistance of low-carbon steel in a chloride containing corrosive medium, with the latter causing general localized corrosion. Both zinc-based alloys were electrodeposited on the low-carbon steel substrate followed by deposition of an additional surface sol-gel coating of $TiO_2$. Both bi-layer systems demonstrate enhanced corrosion resistance and a protective ability compared to the ordinary zinc. For example, their corrosion potentials are more positive with approximately 150–250 mV compared to the ordinary zinc, and their corrosion currents are approximately 5–7 times lower (see Table 2). In addition, at the end of the 35-day immersion test period, the $R_p$ values of both systems are approximately 11–17 times greater compared to the zinc.

Several reasons can be summarized in order to explain the better corrosive resistance of System B (Zn-Co/$TiO_2$) compared to System A (Zn-Ni/$TiO_2$):

- suitable surface morphology, i.e., the surface is more even and uniform, without holes, pits or cracks;
- the system is characterized by a greater degree of amorphousness of the top-layer, which is confirmed by XRD studies;
- the appearance of a newly formed intermetallic compound, $CoZn_{13}$, which demonstrates better corrosion resistance.

The roughness $R_q$ of the Zn-Ni/TiO$_2$ is 49.5 nm, while for Zn-Co/TiO$_2$, the $R_q$ value is 53.4 nm. The water contact angles are 93.2 and 95.5 for the Zn-Ni/TiO$_2$ and Zn-Co/TiO$_2$ systems, respectively.

Barrier effects created by the presence of the newly applied corrosion products and the sol-gel coating seems to contribute to the enhanced protective properties of System B compared to System A and to the ordinary zinc coating.

**Author Contributions:** Conceptualization, I.S., D.S., N.B. (Nelly Boshkova) and N.B. (Nikolai Boshkov); methodology, S.S., N.G., G.A. and M.S.; software, O.D., G.A., S.S. and N.G.; validation, N.B. (Nikolai Boshkov); formal analysis, V.B., M.P. and M.S.; investigation, N.B. (Nelly Boshkova), N.B. (Nikolai Boshkov), D.S. and I.S.; writing—I.S., D.S. and N.B. (Nikolai Boshkov); writing—review and editing, I.S., D.S. and N.B. (Nikolai Boshkov); visualization, O.D., S.S., M.S. and G.A.; supervision, I.S. and N.B. (Nikolai Boshkov); project administration, N.B. (Nelly Boshkova) and N.B. (Nikolai Boshkov). All authors have read and agreed to the published version of the manuscript.

**Funding:** The authors express their gratitude to the project with the Fund "Scientific Investigations", Bulgaria, KP-06-N37/16 (КП-06-Н37/16), "New environmentally friendly one- and multi coatings for corrosion protection of structural materials with wide application" for the financial support and for the possibility to publish the obtained results.

**Institutional Review Board Statement:** Not applicable.

**Informed Consent Statement:** Not applicable.

**Data Availability Statement:** Not applicable.

**Acknowledgments:** The authors express their gratitude to the project with the Fund "Scientific Investigations", Bulgaria, KP-06-N37/16 (КП-06-Н37/16), "New environmentally friendly one- and multi coatings for corrosion protection of structural materials with wide application" for the financial support and for the possibility to publish the obtained results. The support of the European Regional Development Fund within the OP Science and Education for Smart Growth 2014–2020, Project CoE: National Centre for Mechatronics and Clean Technologies, No. BG05M2OP001-1.001-0008 is also acknowledged. Research equipment of the Distributed Research Infrastructure INFRAMAT, part of the Bulgarian National Roadmap for Research Infrastructures, supported by the Bulgarian Ministry of Education and Science, was used in these investigations.

**Conflicts of Interest:** The authors declare no conflict of interest.

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
