# Peer review of "Protective Characteristics of TiO2 Sol-Gel Layer Deposited on Zn-Ni or Zn-Co Substrates"

_coatings, doi:10.3390/coatings13020295_

Round 1

Reviewer 1 Report

Dear Editor,

There are almost sufficient works have been done in this paper with the title of "PROTECTIVE CHARACTERISTICS OF TiO2 SOL-GEL LAYER DEPOSITED ON Zn-Ni OR Zn-Co SUBSTRATES" to achieve a useful product which in my view it is good to have accepted in the journal of coatings. However, there are some parts that needed to be corrected which are sorted below:

1. The structure of the title is not well defined. In my opinion, it should OR be converted to AND.

2 The abstract is incomplete, it should contain all of detailed information about the prominent achievements of the experiments including numeric results.

3. In the introduction section, it should be better to talk about the advantage of sol-gel method. My suggestion is to use these references:

Najafi et al.,Improvement of ZrB2 nanopowder synthesis by sol-gel method via zirconium alkoxide/boric acid precursors, Journal of Sol-Gel Science and Technology, vol.48, 2022

Najafi et al.,Influence of pH and temperature parameters on the sol-gel synthesis process of meso porous ZrC nanopowder,Ceramics International, vol.48, number18, 2022

4. In the results and discussion section, figures 1 and 4 are of poor quality. Please correct it.

5. In the conclusion, the writer should bring their numeric results with their discussions altogether.

6. Unfortunately, I could not find any novelty in this work. Please elucidate it.

Author Response

Dear Editor,

There are almost sufficient works have been done in this paper with the title of "PROTECTIVE CHARACTERISTICS OF TiO2 SOL-GEL LAYER DEPOSITED ON Zn-Ni OR Zn-Co SUBSTRATES" to achieve a useful product which in my view it is good to have accepted in the journal of coatings. However, there are some parts that needed to be corrected which are sorted below:

  1. The structure of the title is not well defined. In my opinion, it should OR be converted to AND.

The comment is taken into consideration and the title is corrected.

2 The abstract is incomplete, it should contain all of detailed information about the prominent achievements of the experiments including numeric results.

The comment is taken into and the needed text is added in the Abstract part.

  1. In the introduction section, it should be better to talk about the advantage of sol-gel method. My suggestion is to use these references:

Najafi et al.,Improvement of ZrB2 nanopowder synthesis by sol-gel method via zirconium alkoxide/boric acid precursors, Journal of Sol-Gel Science and Technology, vol.48, 2022

Najafi et al.,Influence of pH and temperature parameters on the sol-gel synthesis process of meso porous ZrC nanopowder,Ceramics International, vol.48, number18, 2022

The comment is taken into account. Additional text and both references are added to the manuscript.

  1. In the results and discussion section, figures 1 and 4 are of poor quality. Please correct it.

The comment is taken into account and the quality of the figures is improved.

  1. In the conclusion, the writer should bring their numeric results with their discussions altogether.

 The comment is taken into account and the required text is added in the Conclusion part.

  1. Unfortunately, I could not find any novelty in this work. Please elucidate it.

To the best of our knowledge the sol-gel method is applied to ensure better corrosion resistance and protective ability of the steel – low-carbon or alloyed. In the available scientific literature, the information about the obtaining and investigation of the corrosion behavior of protective systems based on low-carbon steel, covered with а zinc alloy (Zn-Ni or Zn-Co) and additionally with second sol-gel titanium dioxide layer is practically missing.

The aim of the present investigation is to characterize the corrosion properties of two novel bi-layer systems containing Zn-Co(3 wt.%) or Zn-Ni(10 wt.%) as under-layers and additional TiO2 sol-gel coating as top-layer in a model corrosive medium of 5% NaCl and to compare it with ordinary zinc coating. Additional aim is to evaluate the impact of the nature of the under-layer.

Reviewer 2 Report

Dear Authors,

* The curves for Zn-Co/TiO2 and Zn-Ni/TiO2 in Figure 12 are drastically different. Is that possible. If so, explain the reasons.

* I think that the English language of the work should be checked and proofread.

Author Response

Dear Authors,

* The curves for Zn-Co/TiO2 and Zn-Ni/TiO2 in Figure 12 are drastically different. Is that possible. If so, explain the reasons.

The comment of the reviewer is right. This result was also surprising to us at the first moment. In our opinion this difference is explained and commented in the Conclusion part. The reasons can be found in the more even and uniform surface morphology, in the greater degree of amorphousness (confirmed by XRD studies) as well as in the appearance of newly formed intermetallic compound CoZn13 which demonstrates better corrosion resistance.

* I think that the English language of the work should be checked and proofread.

 The comment is taken into account and the English text has been improved by the Editor of journal “Nanoscience and Nanotechnology” of the Bulgarian Academy of Sciences ISSN:1313-8995 following the Instructions of “American Chemical Society Style Guide – A Manual for Authors and Editors” Janet S. Dodd, Editor, Marianne C. Brogan, Advisory Editor, ACS Washington DC 1986

Reviewer 3 Report

The article “Protective characteristic of TiO2 Sol-Gel layer deposited on Zn-Ni or Zn-Co substrated” by Nelly Boshkova et al. is well written and the method is well developed and explained. The authors have used different techniques to characterize the protective layer and the products of corrosion, which allows better understanding on the changes of the surface under corrosion.  Few points need to be revised before acceptance for publication.

Please check all abbreviations are well defined before the utilization later in the text.

I think it is better to add the photos of the sample, before and after application of the gel and also the corrosion layer.

All the SEM images are flurry, please change the contrast in order to reveal the important information.

Between Figure 2 and 3, the Ti signal is not different, please check the comments of these results.

Between Figure 5 and 6, the information of carbon is missing, may the authors put some comments?

Figure 11, for the X-axis, check the numbers that need be in the correct format.

In the part of discussion, the authors should compare the different techniques used, and give some comments if all techniques are needed for future analysis if the new products are invented.

And also, for theses two systems presented in the article, how it can prevent the corrosion and how the interaction between this sol-gel layer and the metals.

Author Response

The article “Protective characteristic of TiO2 Sol-Gel layer deposited on Zn-Ni or Zn-Co substrated” by Nelly Boshkova et al. is well written and the method is well developed and explained. The authors have used different techniques to characterize the protective layer and the products of corrosion, which allows better understanding on the changes of the surface under corrosion.  Few points need to be revised before acceptance for publication.

Please check all abbreviations are well defined before the utilization later in the text.

The comment is taken into consideration and the abbreviations are checked.

I think it is better to add the photos of the sample, before and after application of the gel and also the corrosion layer.

Generally, the authors agree with this recommendation. However, in our opinion such additional images would lead to cluttering the article with photographic material, which would make it difficult to perceive its main idea. Оn the other hand, photos of galvanized steel and sol-gel coatings could be seen in articles by our colleagues working on similar topics. These are the reasons why the authors have not attached such images and we hope to be correctly understood by the esteemed reviewer.

All the SEM images are flurry, please change the contrast in order to reveal the important information.

The comment is taken into consideration and the contrast of the SEM images is changed.

Between Figure 2 and 3, the Ti signal is not different, please check the comments of these results.

The reviewer is right that the Ti signal registered in both figures is practically the same. The authors opinion is that the probable reason for this is the better corrosion resistance of TiO2-based sol-gel layer in that test medium.

Between Figure 5 and 6, the information of carbon is missing, may the authors put some comments?

The comment is taken into consideration and the needed information (carbon peak) is added to Figure 6.

Figure 11, for the X-axis, check the numbers that need be in the correct format.

The comment is taken into consideration and the numbers are changed to the correct format.

In the part of discussion, the authors should compare the different techniques used, and give some comments if all techniques are needed for future analysis if the new products are invented.

The comment is taken into consideration and the needed text (marked in green) is added in the Discussion part.

And also, for theses two systems presented in the article, how it can prevent the corrosion and how the interaction between this sol-gel layer and the metals.

The sol-gel method is applied to ensure better corrosion resistance and protective ability of the steel – low-carbon or alloyed. Our preliminary experiments showed that the application of sol-gel method directly on ordinary zinc (galvanized steel substrate) resulted in a coating with bad decorative appearance and lower corrosion resistance due the presence of many flakes and bad adhesion. The novelty is the application of some zinc based alloys aiming to ensure better decorative appearance and higher corrosion resistance in the presence of chloride ions.To the best of our knowledge the information about the obtaining and investigation of the corrosion behavior of protective systems based on low-carbon steel, covered with а zinc alloy (Zn-Ni or Zn-Co) and additionally with second sol-gel titanium dioxide layer is practically missing in the available literature.

The aim of the present investigation is to characterize the corrosion properties of two novel bi-layer systems containing Zn-Co(3 wt.%) or Zn-Ni(10 wt.%) as under-layers and additional TiO2 sol-gel coating as top-layer in a model corrosive medium of 5% NaCl and to compare it with ordinary zinc coating. Additional aim is to evaluate the impact of the nature of the under-layer.

Reviewer 4 Report

In this work, systems of Zn-Co (3 wt.%) or Zn-Ni (10 wt.%) alloy coatings and a very thin TiO2 sol-gel film were thoroughly investigated by providing different experimental studies. The experimental results prove the constructive impact of the proposed systems on the enhanced protective properties of low-carbon steel. The paper is well organized, and the data provided is of significant interest in the research field. Here are some comments to improve the presented work:

1.     Although the introduction covers the subject and the aims in an appropriate way and also includes a reasonable literature survey, it still needs more references that are recently published in 2022 to highlight the state-of-the-art including the possible applications.

2.     Table 2 should be revised

3.     Figure 12: Rp to be corrected to Rp

4.     In the Discussion section, authors are required to include a table to summarize their findings and compare their results with other systems from the literature.

5.     The conclusion should be rewritten to include more quantitative results.

6.     Overall writing skill of this paper is good. However, there are some grammatical mistakes, which must be pointed out by the authors and corrected subsequently.

Author Response

In this work, systems of Zn-Co (3 wt.%) or Zn-Ni (10 wt.%) alloy coatings and a very thin TiO2 sol-gel film were thoroughly investigated by providing different experimental studies. The experimental results prove the constructive impact of the proposed systems on the enhanced protective properties of low-carbon steel. The paper is well organized, and the data provided is of significant interest in the research field. Here are some comments to improve the presented work:

  1. Although the introduction covers the subject and the aims in an appropriate way and also includes a reasonable literature survey, it still needs more references that are recently published in 2022 to highlight the state-of-the-art including the possible applications.

The comment is taken into account and two additional references from the previous year 2022 have been added.

  1. Table 2 should be revised.

The comment is taken into account and Table 2 is revised (last column is removed).

  1. Figure 12: Rp to be corrected to Rp

        The comment is taken into account and the needed correction is done.

  1. In the Discussion section, authors are required to include a table to summarize their findings and compare their results with other systems from the literature.

The comment is taken into account and an additional Table 3 has been added summarizing the results from the polarization resistance measurements. As well known the sol-gel method is applied to ensure better corrosion resistance and protective ability of the steel – low-carbon or alloyed. Our preliminary experiments showed that the application of sol-gel method directly on ordinary zinc (galvanized steel substrate) resulted in a coating with bad decorative appearance and lower corrosion resistance due the presence of many flakes and bad adhesion. The novelty is the application of some zinc based alloys aiming to ensure better decorative appearance and higher corrosion resistance in the presence of chloride ions. Since practically the presence of such data (application of sol-gel method on zinc based alloys) is rather scarce we have compared the obtained experimental data with the ordinary zinc. 

  1. The conclusion should be rewritten to include more quantitative results.

The comment is taken into account and the needed correction is done.

  1. Overall writing skill of this paper is good. However, there are some grammatical mistakes, which must be pointed out by the authors and corrected subsequently.

The comment is taken into account and the English text has been improved by the Editor of Journal “Nanoscience and Nanotechnology” of the Bulgarian Academy of Sciences, ISSN:1313-8995, following the Instructions of “American Chemical Society Style Guide – A Manual for Authors and Editors” Janet S. Dodd, Editor, Marianne C. Brogan, Advisory Editor, ACS Washington DC 1986

Round 2

Reviewer 1 Report

Dear Editor,

The manuscript has been revised carefully and it can be considered for acceptance.

Author Response

Dear Editor,

The manuscript has been revised carefully and it can be considered for acceptance

The authors express their gratitude to the reviewer.

Reviewer 4 Report

The authors have addressed the raised issues from the previous review. The paper has been improved and I recommend it for publication.

Author Response

The authors have addressed the raised issues from the previous review. The paper has been improved and I recommend it for publication.

The authors express their gratitude to the reviewer.